# Role of an ancient light-harvesting protein of PSI in light absorption and photoprotection

Yandu Lu [1,2✉], Qinhua Gan[1], Masakazu Iwai [2,3], Alessandro Alboresi [4], Adrien Burlacot[5], Oliver Dautermann [2], Hiroko Takahashi[6], Thien Crisanto[2], Gilles Peltier [5], Tomas Morosinotto [4], Anastasios Melis [2] & Krishna K. Niyogi [2,3✉]

Diverse algae of the red lineage possess chlorophyll *a*-binding proteins termed LHCR, comprising the PSI light-harvesting system, which represent an ancient antenna form that evolved in red algae and was acquired through secondary endosymbiosis. However, the function and regulation of LHCR complexes remain obscure. Here we describe isolation of a *Nannochloropsis oceanica* LHCR mutant, named *hlr1*, which exhibits a greater tolerance to high-light (HL) stress compared to the wild type. We show that increased tolerance to HL of the mutant can be attributed to alterations in PSI, making it less prone to ROS production, thereby limiting oxidative damage and favoring growth in HL. HLR1 deficiency attenuates PSI light-harvesting capacity and growth of the mutant under light-limiting conditions. We conclude that HLR1, a member of a conserved and broadly distributed clade of LHCR proteins, plays a pivotal role in a dynamic balancing act between photoprotection and efficient light harvesting for photosynthesis.

[1] State Key Laboratory of Marine Resource Utilization in South China Sea, College of Oceanology, Hainan University, Haikou, Hainan, China. [2] Howard Hughes Medical Institute, Department of Plant and Microbial Biology, University of California, Berkeley, CA, USA. [3] Molecular Biophysics and Integrated Bioimaging Division, Lawrence Berkeley National Laboratory, Berkeley, CA, USA. [4] Dipartimento di Biologia, Universita' di Padova, Padua, Italy. [5] CEA, CNRS, Aix-Marseille Université, Institut de Biosciences et Biotechnologies Aix-Marseille, UMR 7265, Laboratoire de Bioénergétique et Biotechnologie des Bactéries et Microalgues, CEA Cadarache, Saint-Paul-lezDurance, France. [6] Department of Biochemistry and Molecular Biology, Graduate school of Science and Engineering, Saitama University, Saitama, Japan. ✉email: ydlu@hainanu.edu.cn; niyogi@berkeley.edu

Marine microalgae constitute an important component of the biological carbon pump[1] and produce approximately half of the oxygen on earth[2]. They are widespread throughout the oceans and exhibit extreme flexibility in changing environmental conditions, which is thought to be a key element that has driven their rise to dominance in contemporary oceans[3]. To cope with various light regimes, microalgae have undergone significant structural and functional divergence of their light-harvesting complexes (LHCs). These intrinsic membrane antenna systems are generally used to harvest sunlight and transfer excitation energy to the reaction centers to drive photosynthesis, but some function in energy dissipation, such as the PSII-associated PsbS (in all land plants)[4], the light-harvesting complex stress-related (LHCSR) proteins of PSII (in Chlamydomonas, other green algae, and mosses)[5], and LHCX1 (in diatoms[6,7] and Nannochloropsis[8]).

From a common origin, which was proposed to be the high-light-inducible proteins (HLIPs) of cyanobacteria[9], LHCs have evolved tremendous diversity in different groups of photosynthetic eukaryotes. Many cyanobacteria rely on phycobiliproteins for light-harvesting, which are organized as phycobilisomes that are peripherally associated with thylakoid membranes. Viridiplantae (including plants and green algae) and red algae evolved after a primary endosymbiosis between a cyanobacterium and a host eukaryote. The green lineage lost the phycobilisome, but acquired the chlorophyll (Chl) a- and Chl b-containing LHCs, specifically associated with the two photosystems, PSI and PSII. In contrast, red algae possess two distinct types of functional peripheral antenna complexes, i.e., Chl a-binding polypeptides (termed LHCR and associated with PSI) and phycobilisomes (the predominant antenna for PSII), representing an evolutionary intermediate between the prokaryotic cyanobacteria and the green lineage[10]. LHCR could thus represent an ancient antenna form in photosynthetic eukaryotes that was maintained by present-day red algae.

The red lineage represents a diverse group of eukaryotic phototrophs derived from one or several secondary endosymbioses involving a red alga[11]. Members of the red lineage include stramenopiles (also called heterokonts), such as Bacillariophyceae (diatoms), Phaeophyceae (brown algae), and Eustigmatophyceae. These algae share quite similar plastids, and based on genome evidence, they possess similar antenna complexes, including LHCRs, suggesting a wide presence and indispensable role(s) of LHCRs in extant microalgae[12,13]. However, knowledge of the structure and function of LHCRs remains elusive[14].

Nannochloropsis spp. are a genus of unicellular photosynthetic microalgae belonging to the eustigmatophytes. Their light-harvesting antennae have a distinct property of binding only Chl a, with no other accessory Chls, whereas Chl c is predominant in the antennae of other stramenopiles[13]. Moreover, different from the abundance of fucoxanthin in diatoms and brown algae, Nannochloropsis contains violaxanthin and vaucheriaxanthin esters as the most abundant carotenoids[14]. Therefore, the photosynthetic apparatus of these algae possess idiosyncratic features with respect to other stramenopiles, and its characterization could contribute to a better understanding of LHC diversity and evolution in photosynthetic organisms.

In this work, an N. oceanica mutant strain is isolated based on its greater tolerance to high light (HL). The gene responsible for this phenotype, named high-light resistance 1 (HLR1), is identified and validated by isolating alternative knockout lines by CRISPR-Cas9 and RNAi approaches that are specifically depleted in HLR1, a 21 kD light-harvesting antenna protein with similarity to LHCR. HLR1 absence has a significant impact on PSI, causing a substantially smaller antenna size and depletion of PSI–LHCI holocomplexes that are normally present in WT cells. The alterations in PSI cause a decrease in reactive oxygen species (ROS) production from PSI and, in particular, superoxide radicals and hydrogen peroxide ($H_2O_2$), explaining the mutant's increased tolerance to HL.

## Results

### The hlr1 mutant maintains a high PSII efficiency in HL.
The hlr1 mutant was obtained by genetic selection of Nannochloropsis oceanica, in which a library of mutants generated by random DNA insertional mutagenesis was subjected to high irradiance (HL, 200 μmol photons m$^{-2}$ s$^{-1}$). Out of 50,000 independent transgenic lines, only six mutants were eventually able to grow on f/2 plates (300 μg L$^{-1}$ hygromycin B) exposed to HL. Among them, one strain (designated as hlr1, high-light resistance 1) had a green coloration under low-light growth conditions, similar to the wild type (WT) (Fig. 1a, LL, 5 μmol photons m$^{-2}$ s$^{-1}$). However, when cultivated under continuous HL, it displayed a green coloration compared with the yellow-green color of the WT (Fig. 1a, HL). Under LL growth conditions, both WT (161.2 ± 0.5 pg cell$^{-1}$) and hlr1 (164.8 ± 0.33 pg cell$^{-1}$) had a similar Chl a content. Growth under high light caused an uneven lowering of the Chl a content in the two strains (Fig. 1b), with the WT measuring 107.7 ± 6.4 pg cell$^{-1}$ and hlr1 measuring 134.1 ± 9.8 pg cell$^{-1}$. Thus, under HL, the hlr1 mutant showed a statistically significant 28% greater Chl a content than that in the WT (Fig. 1b). The hlr1 mutant showed a slightly compromised rate of oxygen evolution (Supplementary Fig. 1a) and growth under LL conditions (Supplementary Fig. 1b). In contrast, the hlr1 mutant showed a moderately higher oxygen evolution than the WT under HL growth conditions (Supplementary Fig. 1a). LL cells exposed to HL conditions showed a decrease in maximum photochemical efficiency of PSII (in terms of $F_v/F_m$ values), presumably due to photoinhibition. This adverse effect was less pronounced in the hlr1 mutant than in the WT, suggesting higher tolerance to strong illumination (Fig. 1c). Prolonged HL induced a massive membrane rearrangement in WT cells with several plastoglobules also visible, while this effect was less evident in the hlr1 mutant, which maintained a relatively similar cell structure as under LL conditions (Fig. 1d and Supplementary Fig. 2).

### The hlr1 mutant is defective in an HL-induced LHCR gene.
A single band was observed by using the eHYG (hygromycin) probe in a DNA gel blot of the hlr1 genomic DNA, revealing a single transgene integration in the N. oceanica genome (Supplementary Fig. 3). The flanking genomic sequence retrieval (Supplementary Data 1) and genome-wide examination (transcriptomics datasets of the hlr1 mutant under darkness (DK) and HL conditions over three time points; see "Methods" for details; Supplementary Data 2) confirmed a single insertion site within the exon region of a gene (number 4928) encoding an LHC protein, which we designated as HLR1 (high-light resistance 1) (Fig. 2a). Cultures dark-acclimated overnight were transferred to 200 μmol photons m$^{-2}$ s$^{-1}$. There was a discernible induction at both the mRNA (Fig. 2b) and protein (Fig. 2c) levels of HLR1 after 24 h of exposure to high irradiance in WT, followed by a lowering of transcript levels between 24 and 48 h after the transfer. The HLR1 protein was present in the WT but not detected in hlr1 cells, under either DK or HL incubation conditions, as evidenced in immunoblot analysis with anti-HLR1 antibodies (Fig. 2c), suggesting a loss of HLR1 protein accumulation in the hlr1 mutant.

To confirm that mutation of HLR1 was responsible for the observed phenotypes, CRISPR/Cas9 knockout lines of HLR1 were generated in N. oceanica strain CCMP 1779 (1779-HLR1$_{KO}$), while RNAi-mediated knockdown strains of HLR1 were generated in N. oceanica strain IMET1 (IMET1-HLR1$_i$) with depressed HLR1 protein levels (Supplementary Fig. 4 inset; see "Methods"

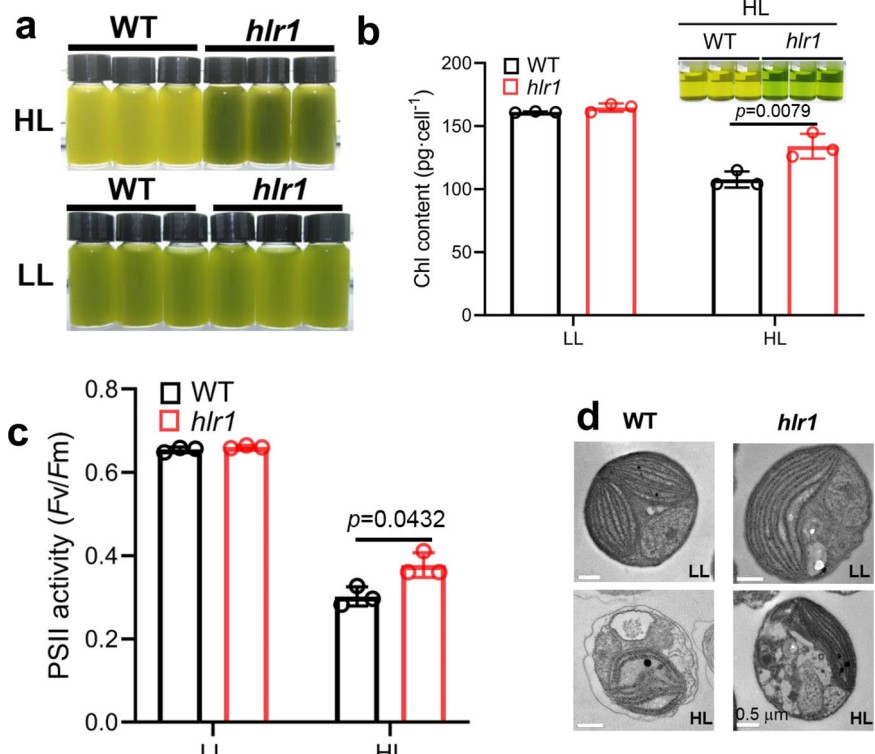

**Fig. 1 The *N. oceanica hlr1* mutant is tolerant to high light. a** Cultures of WT and the *hlr1* mutant grown in LL or HL conditions. Note the dark coloration of the *hlr1* strains as compared with the pale coloration of WT under HL conditions for 8 days. **b** Chl quantification. The inset shows a comparison of pigment extraction of WT and the *hlr1* mutants under HL conditions. **c** PSII photochemical efficiency measurements. Algal cells in the mid-logarithmic growth phase were collected and resuspended to equal cell densities. After overnight acclimation, cells were transferred to the indicated light regime for 24 h, then the maximum PSII photochemical efficiency ($F_v/F_m$) was measured. **d** Transmission electron microscopy images. Bar = 0.5 μm. LL, low light (5 μmol photons m$^{-2}$ s$^{-1}$); HL, high light (200 μmol photons m$^{-2}$ s$^{-1}$). Data are presented as the means ± SD ($n = 3$ for **b**, **c**). The *P* values with significance are shown.

for details of the generation and validation of these mutants). Like the original *hlr1* mutant, the IMET1-HLR1$_i$ (Supplementary Fig. 4) and 1779-HLR1$_{KO}$ strains (Supplementary Fig. 5) were less sensitive to saturating light than their respective WT strains, firmly establishing the unambiguous relationship between high-light resistance and HLR1 deficiency.

Topology prediction of the HLR1 supports a folding model with three transmembrane helices (TM1, TM2, and TM3 in Fig. 2d), a stroma-exposed N-terminus, and a lumen-exposed C-terminus (Fig. 2d). Hidden Markov model-based searches suggest structural similarity of HLR1 to the PSI–LHCI from a green alga *Bryopsis corticulans* (PDB entry 6IGZ in Pfam; http://pfam.xfam.org/structure/6igz)[15]. HLR1 has 162 residues (81%) modeled with 100.0% confidence by this single template. The transmembrane domains and C-terminus are more conserved than the N-terminal alanine-rich domain. Like other LHCs, the TM1 of HLR1 shares sequence similarity to cyanobacterial HLIPs (single transmembrane polypeptide[9]) with conserved Chl *a*-binding sites (Supplementary Fig. 6a), supporting that LHC polypeptides may have arisen by gene duplication of an ancestral gene encoding a single TMH polypeptide[16]. The phylogenetic analysis revealed a high level of sequence conservation between HLR1 and the LHCR-type proteins, previously found in association with PSI in *N. gaditana*[17] and in the red alga *Cyanidioschyzon merolae*[12]. These LHCR sequences form a clearly separate clade with respect to stress-related LHC proteins involved in NPQ activation (e.g., *Chlamydomonas* LHCSR3[5] and LHCX1 in *Phaeodactylum*[7]). Antennae associated with PSII in heterokonts like *Nannochloropsis* (LHCF) are found in another separate clade. The chlorophyll *a/b*-binding LHCA proteins associated with PSI in plants and green algae

clearly form a separate outgroup (Fig. 2e), suggesting that divergence of the different LHC subfamilies occurred independently in the red and green lineages[18]. It is also interesting to observe that HLR1, while belonging to the LHCR clade, forms a statistically significant isolated LHCR5 cluster compared to other LHCR proteins from *Nannochloropsis gaditana* and other algae from the red lineage. This suggests the possibility of functional specialization of this protein with respect to the other LHCRs.

Similar to *C. reinhardtii* LHCSR3, the HLR1 abundance increased upon exposure of cells for a few hours to saturating light conditions, during which many algal species up-regulate energy-dependent quenching (qE) capacity[5]. To explore whether HLR1 is involved in non-photochemical quenching (NPQ), chlorophyll fluorescence quenching and HPLC analysis of carotenoids in response to dark/light exposure were investigated. The *hlr1* mutant has a slightly lower NPQ capacity (Supplementary Fig. 7a) and significantly altered violaxanthin (V; Supplementary Fig. 7b) and antheraxanthin (A; Supplementary Fig. 7c) accumulation in response to high-light exposure. The ability to accumulate zeaxanthin (Z; Supplementary Fig. 7d) was compromised in the *hlr1* mutant and the value of $(A + Z)/(V + A + Z)$ was lower in the mutant than that of WT (Supplementary Fig. 7e). However, the pH-sensing sites in *Chlamydomonas* LHCSR, essential for triggering NPQ (spheres in Supplementary Fig. 6b)[19], are absent in HLR1. Moreover, in contrast to *LHCSR or LHCX* mutants, acclimation to saturating illumination (200 μmol photons m$^{-2}$ s$^{-1}$) for 24 h induced stronger NPQ activity in the *hlr1* cells than WT (Supplementary Fig. 7f). Overall, while the HLR1 defect affects the NPQ capacity of the mutant, its influence on NPQ appears to be indirect and different

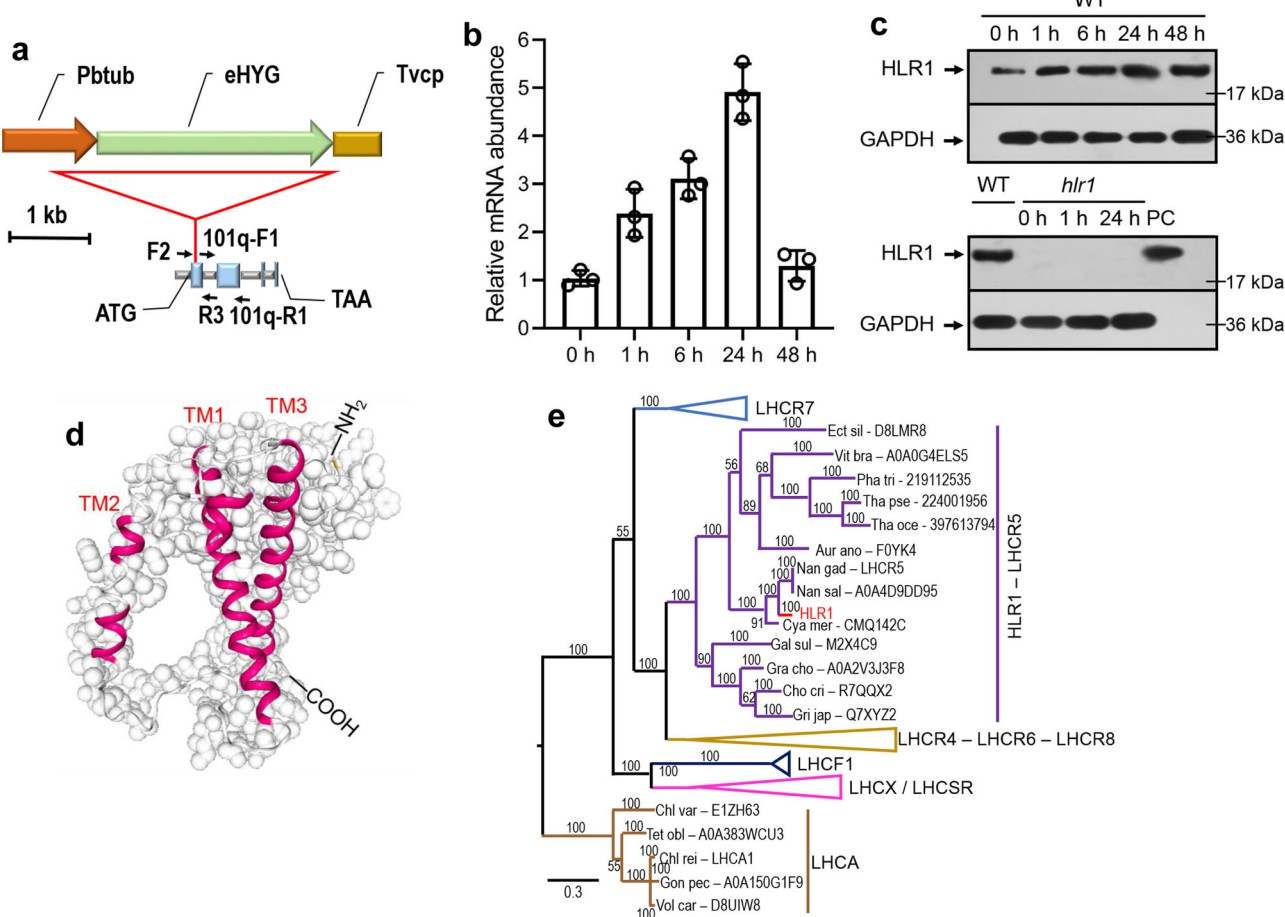

**Fig. 2 Molecular and biochemical analysis of HLR1. a** A schematic representation of the pMEM2 cassette insertion site in the genome of the *hlr1* mutant. The red-line triangle represents the insert. Thick arrows indicate the orientation of the positive strand of the *eHYG* gene conferring resistance to hygromycin B. The *HLR1* gene map is shown in a 5′ to 3′ direction from left to right, with exons and introns represented by filled boxes and connecting lines, respectively. Arrows indicate the primers used for qPCR (101q-F1 and 101q-R1) and amplifying regions across the ends of the insertion and the flanking sequences (F2 and R3). **b** Quantitative RT-PCR of the *HLR1* transcript in wild-type parental strain upon high irradiance. The primers 101q-F1 and 101q-R1 were used for qPCR. Data are presented as means ± SD (*n* = 3). All primers are listed in Supplementary Data 5. **c** Immunoblot analysis of HLR1 protein expression in WT and *hlr1*. Protein samples from the *hlr1* mutant were taken from dark-acclimated cells or following high irradiance for the indicated time. Antibodies against glyceraldehyde 3-phosphate dehydrogenase (GAPDH) were used as a loading control. PC, positive controls (recombinant HLR1 proteins expressed in *E. coli*). **d** The folding model of an HLR1 monomer with a view from a direction parallel to the membrane plane. Letters in red show the transmembrane helices. **e** Phylogenetic analysis of light-harvesting complex (LHC) proteins. Four subgroups were collapsed (LHCR7, LHCR4–LHCR6–LHCR8, LHCF1, and LHCX/LHCSR). HLR1 – LHCR5 subgroup was not collapsed. LHCA sequences were chosen as outgroup. Scale bar, 0.3 estimated substitutions per site. See Supplementary Data 6 for the full protein list obtained from the Uniprot and PhycoCosm databases.

from that in the *Chlamydomonas* LHCSR3[5], *Phaeodactylum* LHCX[7], or *N. oceanica* LHCX1[8].

**HLR1 is part of the PSI antenna**. To test the association of the HLR1 protein with PSI or PSII, fractionation of alpha-dodecylmaltoside (α-DM)-solubilized thylakoids from WT and the *hlr1* mutant was performed by blue-native polyacrylamide gel electrophoresis (BN-PAGE) (Fig. 3a). Several bands were resolved in the BN-PAGE gel, whose identities were confirmed by immunoblot analysis with anti-PsaA antibody (a marker for the PSI reaction center), anti-D1 antibody (a marker for the PSII reaction center), and anti-Cyt $b_6$ antibody (a marker for the inter-photosystem electron transport chain) (Fig. 3a). No significant differences were observed between the two genotypes (WT and *hlr1*) in the lower portion of the gel containing complexes <720 kDa, including LHC monomers, PSII core complexes, and Cyt $b_6f$ complexes (Fig. 3a). The upper portion of the gel, resolving high molecular weight supercomplexes, however, showed clear differences (Fig. 3a). A band

with an apparent molecular weight of ~ 1000 kDa (B1 in Fig. 3a) was absent from the *hlr1* samples, which could be attributed to the absence of the largest PSI–LHCI supercomplexes in the mutant, evidenced by immunoblot analysis (Fig. 3a).

To assess the polypeptide composition of the largest PSI–LHCI supercomplexes, the WT band from the BN-PAGE (B1 in Fig. 3a) was excised and subjected to in-gel tryptic digestion followed by identification by mass spectrometry. This band contained HLR1, PsaA, PsaB, PsaC, ferredoxin components, ATPase and ATP synthase, LHCs, and some unique peptides involved in electron transfer and cell redox homeostasis (Supplementary Data 3). In order to account for co-migration of PSI–LHCI supercomplexes, proteins were also eluted and analyzed from an equivalent gel region in the lane from the *hlr1* samples (B2 in Fig. 3a) where HLR1 was not detected (Supplementary Data 3 and Fig. 3a).

To determine whether HLR1 impacts the functional light-harvesting antenna size of the photosystems, we measured the absorption cross-section σ of PSII from the kinetics of the variable fluorescence induction in the presence of DCMU (Fig. 3b,

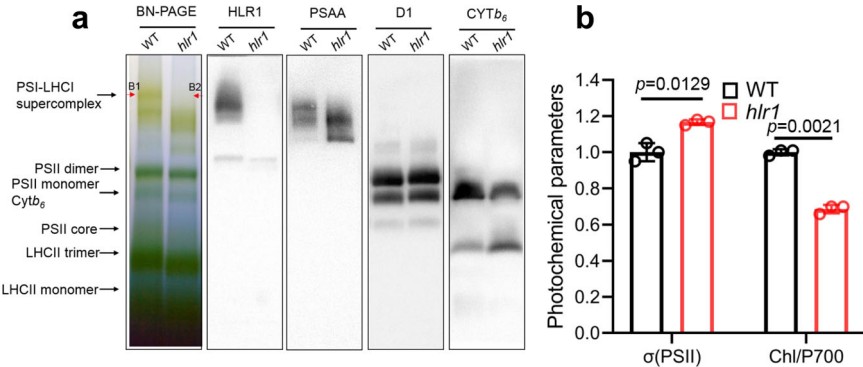

**Fig. 3 Effects of HLR1 defect on the organization of the photosystems in WT and the *hlr1* mutant. a** BN-PAGE and immunoblot analysis of supernatant from solubilized thylakoid membranes from WT and the *hlr1* mutant. Protein complexes were identified with appropriate antibodies. The excised bands from the BN-PAGE for mass spectrometry analysis are indicated as B1 (of WT) and B2 (of *hlr1*). **b** Spectrophotometric measurement of the absorption cross-section of PSII and the relative Chl and PSI content of WT and the *hlr1* mutant. Data are presented as means ± SD ($n = 3$). The *P* values with significance are shown.

σ(PSII)) and the total sample Chl-to-P700 ratio in the WT and *hlr1* mutant. The absorption cross-section σ(PSII) was moderately larger (about 17%) in the mutant compared to the WT (Fig. 3b), affording evidence that the PSII functional antenna size was not largely affected by the mutation. On the other hand, the ratio of total sample Chl-to-P700, measured from Chl *a* quantification and from the light-induced absorbance change at 700 nm, resulted in a Chl/700 949.72 ± 15.77 mol:mol ratio for the WT and 653.36 ± 19.98 mol:mol for the mutant, thus showing a 31% lower Chl content per P700 (per electron transport chain) in the mutant (Fig. 3b). These results suggest a smaller PSI Chl antenna size in the *hlr1* mutant. These immunoblot analysis results corroborate the spectrophotometry analysis and suggest that both the antenna size and the amount of PSI were negatively impacted by the *hlr1* mutation, and they further confirm the association with and a role for the HLR1 in the assembly and function of PSI. A similar negative impact in the amount of chloroplast PSI was seen in the LTD-deficient *C. reinhardtii* mutant, in which the peripheral LHCI import and PSI–LHCI holocomplex assembly were impeded[20].

**HLR1 deficiency impacts electron transport.** The parameter 1-qL $[1-(F_q'/F_v')/(F_o'/F')]$ is considered to be an indicator of the proportion of closed reaction centers and of the "excitation pressure" parameter at PSII[21]. The *hlr1* mutant showed slightly lower values of 1-qL compared with the WT (Fig. 4a), an observation consistent with a lower PSI content. Relative to WT (23.59 ± 2.3), the linear electron flow (LEF) was not significantly different in the mutant (22.61 ± 2.3 e$^-$ s$^{-1}$). In contrast, although at a low level, the cyclic electron transfer (CET) decreased by 29.30 ± 1.9% in *hlr1* (1.16 ± 0.19 e$^-$ s$^{-1}$) compared with WT (1.64 ± 0.23 e$^-$ s$^{-1}$). We also observed a decreased P700$^+$ re-reduction rate in the *hlr1* mutant (Fig. 4b).

Oxygen (O$_2$) could serve as an electron acceptor from PSI and generate reactive oxygen species (ROS), such as superoxide anion radical (O$_2^-$) or hydrogen peroxide (H$_2$O$_2$) (as in the Mehler reaction[22]). We next determined the H$_2$O$_2$ production, which might be affected by the altered electron transport in the *hlr1* mutant. The production of cellular H$_2$O$_2$ was significantly lower in the *hlr1* line than in the WT (Fig. 4c). Superoxide dismutase (SOD) converts the O$_2^-$ to H$_2$O$_2$, which is converted to water and oxygen by catalase (CAT) and peroxidase. Under HL conditions, the addition of CAT and SOD simultaneously or CAT (but not SOD) individually decreased the magnitude of PSII photoinhibition in WT, whereas no apparent effect was observed in the *hlr1* cells (Supplementary Fig. 8), indicating a

difference in ROS production in WT and the *hlr1* mutant. Moreover, the alleviation of cellular damage upon application of CAT, but not SOD, suggests that the detoxification of H$_2$O$_2$ is the critical step to mitigate damage to cells. On the other hand, following prolonged HL treatment, relatively higher levels of carotenoids in the *hlr1* mutant (compared with WT) were observed (Supplementary Fig. 9). It might contribute to the increased efficiency of ROS scavenging, together with which the compromised biosynthesis of ROS could jointly lead to the less production of ROS in the mutant in the long-term HL conditions. Furthermore, compared with WT, the *hlr1* mutant was more sensitive to methyl viologen (MV, an efficient electron acceptor at the PSI acceptor side, which is subsequently re-oxidized by transfer of its electrons to oxygen, forming the superoxide anion[23]), but not 2,5-dibromo-6-isopropyl-3-methyl-1,4-benzoquinone (DBMIB, a photosynthetic electron flow inhibitor acting at the Q$_o$ site of the cytochrome *b$_6$f* complex) (Fig. 4d). These results are consistent with the notion of a defect in the *hlr1* strain that impacts photosynthetic electron transport and alters the generation of ROS on the reducing side of PSI.

**Roles of HLR1 in protection from the HL stress.** Global gene expression of WT and the *hlr1* mutant, as a function of time in dark incubation (DK) and under high irradiance was measured by mRNA-Seq. Cultures were maintained in the dark overnight and then exposed to 200 μmol photons m$^{-2}$ s$^{-1}$ for various times. Aliquots of cells for transcript analysis were collected immediately before the light shift, and then 1 and 6 h following the light shift (see "Methods" for details). Following the transition from darkness to HL, transcripts of LHCXs (e.g., *g903* and *g1546*; Supplementary Fig. 10a) and LHCRs (e.g., *g5629*, *g6113*, *g6882*, and *g9713*; Supplementary Fig. 10b) tend to be increased. Under DK conditions, relative to WT, a low number of genes were differentially expressed (*P* value <0.05 and fold change >2) in the *hlr1* cells (only five genes were downregulated; Supplementary Data 4), suggesting that the *hlr1* transcriptome was similar to WT in the absence of light perturbation. Upon high irradiance, the number of differentially expressed genes gradually increased, representing the core of genes progressively activated and gradually turned on/off by increasing light stress, respectively. In total, 30 genes (0.3% of total) were altered in the *hlr1* mutant compared with WT over two time points under high irradiance (Supplementary Data 4). It is interesting to note that the genes involved in photosynthesis did not show any particular differential expression.

Compared with the WT, the L-ascorbic acid (AA) biosynthesis pathway appeared to be decreased in the mutant in several ways:

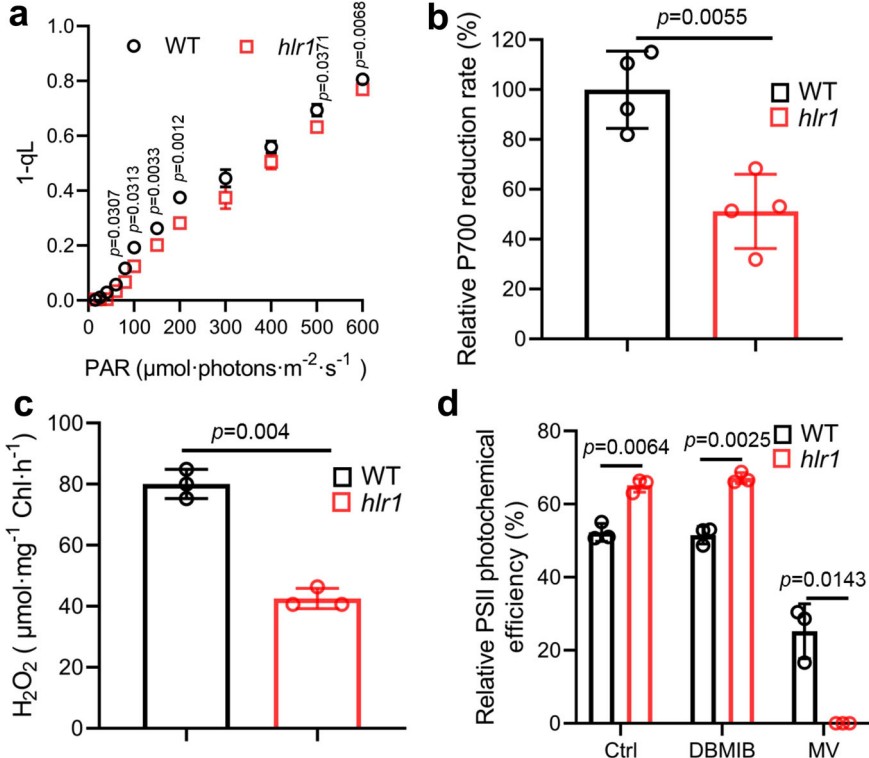

**Fig. 4 Altered electron transport in the WT and the *hlr1* mutant. a** Redox state of the PQ pool (1-qL). **b** Relative P700+ re-reduction rate. **c** Net production of $H_2O_2$ during 1 h high-light treatment. **d** Effects of MV and DBMIB on HL-induced PSII photoinhibition. Cells were incubated in HL for 12 h in the presence or absence of 1 μM DBMIB or 1 mM MV. Data are presented as the means ± SD ($n = 3$ for **a**, **c**, and **d**; $n = 4$ for **b**). The P values with significance are shown.

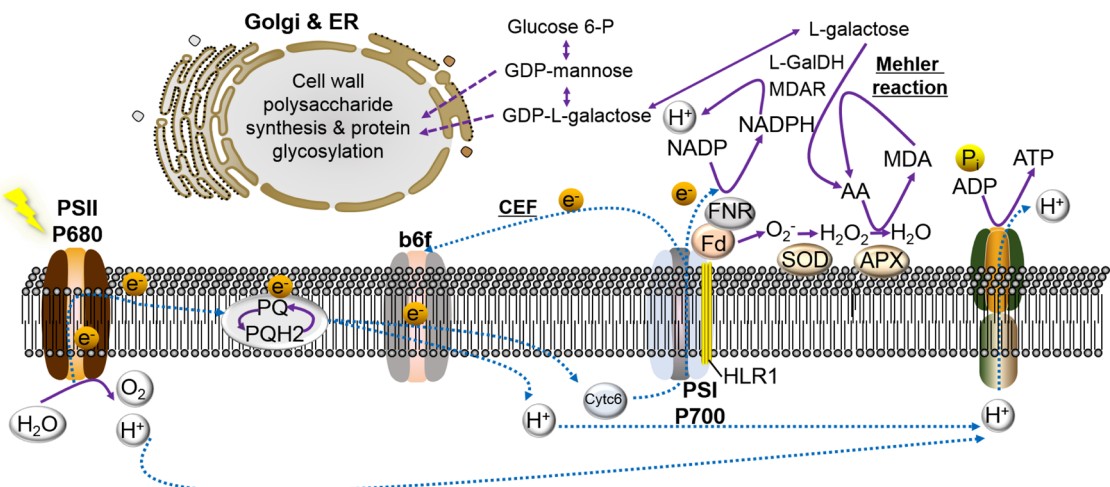

**Fig. 5 Schematic of electron flow in *N. oceanica*.** AA ascorbic acid, APX ascorbate peroxidase, $b_6f$ cytochrome $b_6f$ complex, $Cyt_C6$ cytochrome $c_6$ complex, Fd ferredoxin, FNR ferredoxin-NADP reductase, L-GalDH L-galactose dehydrogenase, MDA monodehydroascorbate, MDAR monodehydroascorbate reductase, PQ and PQH2 plastoquinone and reduced plastoquinone, PSII and PSI photosystem II and I, SOD superoxide dismutase. See Supplementary Fig 10 and Supplementary Data 7 for details of the transcriptional dynamics of genes involved in cell wall biosynthesis, protein glycosylation, and AA biosynthesis.

(i) transcripts of the AA biosynthetic gene L-galactose dehydrogenase[24,25] (L-GalDH, *g10376*) were downregulated (L-GalDH is feedback regulated by AA at the RNA level in some organisms[26]) (Supplementary Fig. 10c); (ii) AA is derived from aldoses such as D-mannose and L-galactose, which can follow several pathways, either toward the biosynthesis of AA, or toward the supply of precursors for cell wall polysaccharide synthesis and protein glycosylation[27] (Fig. 5). HL suppressed the transcripts of

the cellulose biosynthesis gene (i.e., *g7046*) (Supplementary Fig. 10d) and a gene involved in protein deglycosylation (i.e., *g9910*) (Supplementary Fig. 10e).

## Discussion

Microalgae are among the most important and diverse organisms on the planet[28]. They have evolved a broad genomic diversity, which shapes complex phenotypic plasticity and enables them to

adapt and acclimate to diverse environments and ecological niches[29], ranging from the surface to the depth of oceans, and in both terrestrial and aquatic environments. Microalgae, like all photosynthetic organisms, must attain an optimal balance between maximizing sunlight use efficiency and maintaining effective protection from eventual excess of absorbed light energy or excited electrons. Keeping such a balance is particularly challenging in variable environments where energy availability in the form of sunlight and its use via carbon fixation and metabolism, in general, are influenced by many continuously changing parameters (e.g., solar insolation, prevailing temperature conditions, and nutrient availability).

In this study, an *N. oceanica* mutant strain with higher tolerance to saturating illumination was isolated and named *hlr1*. The mutant showed lower susceptibility to PSII photoinhibition (higher $F_v/F_m$ under stress) and an increased rate of oxygen evolution than WT under intense illumination. Multiple independent genetic analyses demonstrated that this increased tolerance was caused by the inactivation of the *HLR1* gene. Based on sequence analysis, HLR1 was identified to be a light-harvesting antenna protein with similarity to LHCR, a group of LHC proteins found in association with PSI in heterokonts and red algae[17]. This identification was confirmed by biochemical data, which showed altered PSI-LHC supercomplex organization and reduced PSI antenna size in the *hlr1* mutant. Thus, our analysis of the *hlr1* mutant has revealed a critical role of a specific clade of LHCR proteins in PSI assembly and function. Using log-phase microalgae, we observed a moderately larger absorption cross-section σ(PSII) (~17%) in the mutant compared to the WT, suggesting a higher PSII antenna size. Meanwhile, the mutant showed lower Chl content per P700 (per electron transport chain) than WT. Overall, in these cells, no significant difference could be observed in Chl content between WT and the mutant. On the other hand, under the high-light condition, the *hlr1* mutant exhibits greater Chl content and a higher maximal quantum yield of PSII, affording evidence that the Chl content in PSII might be higher in the mutant. Thus, higher Chl content in PSII might be a contributor to the larger total Chl content in the *hlr1* mutant in HL conditions.

Oxidative damage of *N. oceanica* cells exposed to high light could be mitigated by treatment with SOD and CAT, which allowed scavenging of the ROS superoxide and $H_2O_2$. This clearly suggests that ROS molecules, mainly produced at the level of PSI[30–32], have a major negative impact on *Nannochloropsis*, when cells are exposed to strong illumination. The ROS generated on the reducing side of PSI could, as a secondary phenomenon, induce the observed PSII photoinhibition, as seen in other species[30–32]. In *hrl1*, the adverse effects are attenuated, consistent with the notion that this mutant generates less ROS at the level of PSI than the WT. This is also corroborated by RNA-Seq analysis that showed, in *hlr1*, lower activation of genes involved in the ROS response, namely ascorbic acid biosynthesis[33,34]. Taken together, all results suggest that the increased tolerance of *hlr1* to HL is attributable to alterations in PSI antenna size and abundance, resulting in lower levels of ROS production, limiting oxidative damage and resulting in better growth. A likely mechanistic explanation is that a smaller light-harvesting antenna size in the *hlr1* mutant results in less excitation pressure to PSI, slower electron transport through this photosystem and, hence, a lower likelihood of ROS production at saturating light intensities.

Although the presence of HLR1 could be harmful to marine microalgae under strong illumination, its conservation in different species suggests that it is essential for competitive survival in the oceanic environment, where sunlight intensity is often well below the saturation point. Indeed, deficiency in HLR1 and the resulting truncated PSI antenna size caused a diminished rate of oxygen evolution and growth under limiting light, suggesting that the absence of the antenna is likely detrimental under those conditions. Thus, the physiological function of HLR1 is to improve sunlight harvesting in low-light conditions, where the main electron sink reactions involve photosynthetic carbon reduction. A truncated light-harvesting antenna size would lead to slower rates of oxygen evolution and a smaller proton gradient across the thylakoid membrane (ΔpH), thereby generating less ATP and leading to compromised growth in limiting light. It would be reasonable to postulate that while *N. oceanica* has a cosmopolitan distribution, it probably does not adapt to strong illumination. Natural selection could have refined its photosynthetic apparatus (e.g., PSI–LHCI supercomplexes) with an emphasis on high efficiency in capturing sunlight, even if this causes an increased sensitivity to photooxidative stress[35]. Thus, this optimized HLR1-dependent light-absorbing capacity appears to be favorable for *N. oceanica* when light intensity is low.

The picture emerging in *N. oceanica* has similarities with some marine cyanobacteria that show a trimeric PSI and several light-harvesting assemblies to cope with limiting light conditions[36]. On the contrary, the observation that PSI is the main target for photoinhibition in *Nannochloropsis* is highly different from plants, where PSI is stable to even very strong illumination. Indeed, PSI antenna mutants in plants have not been demonstrated to be more resistant to strong illumination[32]. In *Nannochloropsis*, PSI is instead sensitive to light stress, consistent with the absence of protective mechanisms such as flavodiiron proteins and the limited capacity of cyclic electron transport[37]. As a consequence in plants, and due to the reported PSI stability, photodamage is primarily limited to PSII, which is efficiently repaired, via the unique D1/32 kD reaction center protein turnover mechanism in the photosynthetic apparatus[38]. While this strategy is convenient because it limits all damage to a single complex that is efficiently repaired, it also heavily relies on PSII reaction center turnover with a consequent high demand for nutrients and nitrogen in particular. Such a protection strategy could thus not be as effective in marine algae-like *Nannochloropsis* that are often exposed to nutrient-limiting environments.

The ability of *Nannochloropsis* to accumulate large amounts of lipids supports the hypothesis that these species are naturally exposed to conditions in which nutrient supply is variable and, in case of limitation, metabolism is directed toward the synthesis of hydrocarbons. Serving to enhance the capture of solar energy when illumination is low (a frequent occurrence in oceans), the HLR1-type antenna may represent an adaptation of these microalgae to life in the oceans, where there is a scarcity of nutrients, in particular iron and nitrogen, and the occasional cell exposure to the surface of the ocean, where higher irradiance conditions prevail. Thus, the regulation of *N. oceanica* photosynthesis, at least from the point of view of the HLR1 protein, could be seen as a dynamic balancing act in which photoprotection is reversibly traded for photosynthetic efficiency.

## Methods

**Growth conditions and microscopy.** *Nannochloropsis oceanica* strain IMET1 was maintained in dim light at 4 °C on solid modified f/2 medium, which was prepared with 35 g L$^{-1}$ sea salt, 1 g L$^{-1}$ NaNO$_3$, 67 mg L$^{-1}$ NaH$_2$PO$_4$·H$_2$O, 3.65 mg L$^{-1}$ FeCl$_3$·6H$_2$O, 4.37 mg L$^{-1}$ Na$_2$EDTA·2H$_2$O, trace metal mix (0.0196 mg L$^{-1}$ CuSO$_4$·5H$_2$O, 0.0126 mg L$^{-1}$ NaMoO$_4$·2H$_2$O, 0.044 mg L$^{-1}$ ZnSO$_4$·7H$_2$O, 0.01 mg L$^{-1}$ CoCl$_2$, and 0.36 mg L$^{-1}$ MnCl$_2$·4H$_2$O), and vitamin mix (2.5 μg L$^{-1}$ VB$_{12}$, 2.5 μg L$^{-1}$ biotin, and 0.5 μg L$^{-1}$ thiamine HCl)[28]. For inoculation, cells were picked into the liquid medium and were maintained under illumination at 50 μmol photons m$^{-2}$ s$^{-1}$ at 25 °C with a light/dark cycle (12 h/12 h). For growth assessment under different light intensities, mid-logarithmic phase algal cells were collected and equal numbers of cells were re-inoculated in liquid media at 25 °C with continuous illumination at 5, 50, or 200 μmol photons m$^{-2}$ s$^{-1}$. For microscopic analysis, cells at indicated periods were sampled for transmission electron microscopy analysis[39].

**Design of plasmids for insertional gene disruption.** To generate a library of *Nannochloropsis* mutants, we employed random insertional mutagenesis with a DNA cassette conferring hygromycin B resistance. The hygromycin resistance (HygR) gene was synthesized to conform to the codon usage in IMET1[40]. The codon-optimized HygR gene was designated as eHYG and was driven by the β-tubulin promoter region (Pbtub) and terminated by the violaxanthin/chlorophyll *a*-binding protein terminator (Tvcp) (pMEM02; Fig. 2a). These individual *cis* elements on the vector were selected or designed based on the IMET1 genome[40]. A mutagenesis library was created by genomic insertion of the *HygR* expression cassette. The proper in vivo functioning of the transforming cassette in IMET1 was verified by selecting transformants on f/2 plates supplemented with 300 µg ml$^{-1}$ hygromycin B. The recognition sites of the Type IIS restriction enzyme *Mme*I were designed at both ends of the transforming cassette. Genomic DNA from mutants was digested with *Mme*I to yield fragments containing the ends of the cassettes and 20 to 21 bp of flanking genomic DNA, which facilitated the retrieval of insertion sites[41]. To identify the locus disrupted by insertional mutagenesis, the flanking sequence was retrieved as described[41] with primers designed for the pMEM02 plasmid (Supplementary Data 5). In brief, ADP1 and ADP2 were designed for an adaptor, which can be ligated to the genomic DNA digested by *Mme*I. The primers used for finding the insertion in the *hlr1* cells were AP1 and 5B1. In addition, nested primers AP2 and 5B2 were used for a second-round PCR. To confirm the results of nested PCR, standard PCR was performed using primers amplifying regions across the ends of the insertion and the flanking sequences (101gi-F2 and 101gi-R3).

**HLR1 phylogeny and topology predictions.** Phylogenetic analysis of the *N. oceanica* HLR1 was performed using amino acid sequences obtained from the Uniprot and PhycoCosm databases. Proteins from all genomes were blasted with the HLR1 protein queries using the standalone BLAST software package. The sequences that were ultimately selected are listed in the Supplementary Data 6. Gaps and ambiguously aligned sites were removed using CLUSTALW 2.0[42], gBlock[43], and manual validation. ProtTest 3[44] and PhyML 3.0[45] were used to select the available model of protein substitution and phylogenetic analyses with a maximum likelihood method. Bootstrap support values were estimated using 100 pseudo-replicates. The three-dimensional (3D) structure was analyzed bioinformatically using the tools PredictProtein[46], Swiss-model[47] and Protein Homology/analogY Recognition Engine V 2.0 (Phyre)[48].

**Phenotyping assays.** The growth of microalgae was monitored by measuring the turbidity or cell number at indicated intervals with a Gene Quant 1300 Spectrophotometer (GE) or a hemocytometer. Dry cell weight of a 10-ml culture was determined simultaneously. For pigment determination, 1 ml algal culture was centrifuged (12,000 × *g* for 3 min) and the supernatant was disposed. Cell pellets were resuspended in 1 ml methanol, disrupted by bead-beating with glass beads for 1 min twice, and left in the dark at 60 °C for 15 min. Then the mixture was centrifuged at 12,000 × *g* for 15 min to remove cellular debris, and the supernatant was used to determine the pigment contents spectroscopically[49].

The kinetics of changes in zeaxanthin concentration was determined by HPLC measurements, as described previously[8]. In brief, log growth-phase cells were dark-acclimated for 30 min, then incubated under 200 µmol photons m$^{-2}$ s$^{-1}$ illumination for variable periods of time, followed by a 5-min dark recovery phase. Aliquots of 500 µL for HPLC analysis were frozen in liquid nitrogen after dark acclimation, 5, 10, and 15 min of illumination, and 5 min of dark recovery. For pigment extraction, the cells were frozen and thawed twice. Cells were broken (6.5 m s$^{-1}$ for 60 s) using the cryorotor of a Fastprep-24 (MP Bio) and extracted with 500 µL of acetone by vortexing. Cell debris was pelleted at 21,000 × *g* for 15 min. Extracted pigments were analyzed with a Spherisorb 5 µm ODS1 column (Waters Corp, USA). More than three biological replicates were examined for each treatment.

**DNA gel blotting and gene expression analysis.** The *N. oceanica HLR1* gene structure was validated by PCR from cDNA and genomic DNA (primers gNoHLR1-F and gNoHLR1-R; Supplementary Data 5). Integration of the transforming cassettes within the genome was analyzed by DNA gel blotting using nonradioactive DIG-containing probes generated by PCR (PCR DIG probe synthesis kit; Roche Diagnostics). Genomic DNA was extracted from microalgal cells using the Plant Genomic DNA Extraction Kit (Omega Bio-tek, Norcross, GA, USA). Ten micrograms of DNA were digested by *Hind*III or *Pst*I and separated on 1% agarose gels, followed by a transfer to membranes. DNA gel blots were then hybridized, washed, and developed.

To produce cDNA templates used for qPCR, the total RNA was purified with the Eastep Super Total RNA Extraction Kit (Promega) and then subjected to genomic DNA digestion and reverse transcription by using PrimeScript RT reagent Kit with gDNA Eraser (Takara). qPCR was performed on a QuantStudio 6 (ThermoFisher Scientific) using SYBR Green PCR Master Mix (Takara) with an equivalent cDNA template and 0.25 µM of each primer. The amount of cDNA input was optimized after serial dilutions. In all qPCR experiments, expression of the target gene was normalized to the expression of the endogenous reference gene actin (g3056), a gene commonly used for normalization in *N. oceanica*[39,50], using the cycle threshold(CT) $2^{-\Delta\Delta CT}$ method. All experiments were done using at least

three biological replicates, and each reaction was run with technical replicates. All primers are provided in Supplementary Data 5.

**Gene silencing and Cas9-mediated gene knockout.** For RNAi vector construction, a 429-bp long fragment (corresponding to the *HLR1* nucleotide sequence from 3367 to 3795) and a 282-bp short fragment (corresponding to the *HLR1* gene sequence from 3367 to 3648 bp) were amplified from the *N. oceanica* cDNA, respectively, with primers NoHLR1i-F, NoHLR1i-R, rcNoHLR1i-F, and rcNoHLR1i-R (Supplementary Data 5). The fragments were ligated in sense and antisense orientations into the *Spe*I and *Sac*II sites of the linearized pMEM03 vector digested by *Spe*I and *Sac*II.

Cas9-mediated target gene disruption was conducted using an episomal CRISPR system[51]. The sgRNAs were designed to meet the following criteria: (1) uniqueness of target sequence and PAM (final 12 bp + NGG) in genome, (2) location within the first 300 bp of the gene, and (3) target sequence present in both genome and mRNA sequence (to avoid targeting introns). Two sgRNAs were synthesized targeting gene 591780 encoding HLR1 in *N. oceanica* CCMP 1779 (Supplementary Data 5). These sgRNAs were ligated into pNOCARS-CRISPR vector and transformed into *N. oceanica* CCMP 1779. Hygromycin was used as antibiotic selection. Mutations with target disruption were screened by PCR with primers HLR1gR-F2 and HLR1gR-R2. The expected products with a size of 1079 bp were extracted, purified, and validated by sequencing.

For transformation, 100 ml wild-type cell cultures were grown to log phase and harvested by centrifugation at 5000 × *g* for 5 min. For each transformation reaction, 10$^9$ cells were mixed with 1 µg cassette in an electroporation cuvette (2 mm gap). Electroporation was performed using the GenePulse Xcell™ (Bio-Rad) with 11 kV cm$^{-1}$ field strength. After the pulse, the cells were immediately mixed with 5 ml f/2 medium for recovery. Then, the cells were plated on the f/2 agar plate with appropriate antibiotic selection until transformant colonies appeared[52].

**Photosynthesis measurements.** PSII photochemical efficiency (measured from the $F_v/F_m$ ratio), NPQ, and qL were monitored through in vivo chlorophyll fluorescence determination with the Imaging PAM (Heinz-Walz)[53] or Fluorescence Monitoring System (Hansatech Instruments). For PSII photochemical efficiency measurements, algal cells in mid-logarithmic growth phase were collected and resuspended to equal cell densities. After overnight acclimation, cells were transferred to the indicated light regime for 24 h, then the $F_v/F_m$ values were measured. Photosystem light-harvesting antenna size was measured using the spectrophotometric kinetic method according to previous reports[54,55]. In brief, the amplitude of the light-minus-dark absorbance difference signal at 700 nm (P700) provided information on the abundance of PSI in wild-type and *hlr1* mutant[55]. The kinetics of $Q_A$ photoreduction of DCMU-poisoned thylakoid were measured, and the corresponding rate constants were used to estimate the functional light-harvesting Chl antenna size of PSII. The ratio of chlorophyll *a* to P700 was also measured in wild type and *hlr1* mutant and used to extract information about the PSI Chl antenna size[55]. Oxygen evolution rates were measured on a standard Clark-type electrode (Chlorolab 2 System; Hansatech Instruments). Experiments were performed using 1.8 ml of microalgal culture at 6 × 10$^7$ cells ml$^{-1}$ in the presence of 0.2 ml 10 mM NaHCO$_3$. Electron transport rates were measured in vivo on log growth-phase microalgae by using a JTS-10 spectrophotometer (Bio-logic, France). The electrochromic band-shift spectral changes were used to calculate electron transport rates[56].

**Thylakoid membrane isolation and blue-native PAGE.** Algal cells were broken twice using a Cell Disruptor (Constant System Cell Disruptor) with a pressure of 20 kPSI (1350 bar). Thylakoid membranes were purified by sucrose cushion centrifugation as described previously[57]. The thylakoids in the appropriate buffer were immediately frozen in liquid nitrogen and stored at −80 °C until further use. Care was taken to avoid repeated freezing and thawing of the thylakoid membrane samples, and the sample preparation for native gel electrophoresis was performed rapidly under dim light at 4 °C. Aliquots of 5 µg Chl of thylakoid membrane samples were solubilized by 2% β-dodecylmaltoside and subsequently separated by blue-native PAGE. Polyclonal antibodies for PsaA (AS06172), D1 (AS10704), and Cyt *b$_6$* (AS184169) (Agrisera, Sweden) were used to identify components in the various electrophoresis bands by immunoblot analysis. The polypeptide composition in PAGE bands of interest was determined upon band excision and in-gel tryptic digestion, followed by identification by mass spectrometry.

**Production of HLR1 antibodies and immunodetection.** HLR1 antibodies were prepared as previously described[23] with some modifications. The topological structure of NoHLR1 was predicted by multiple algorithms (DAS, TMpred, and TMHMM). The hydrophobic transmembrane regions at the N terminal (M1-A23) and the C terminal (G177-S200) were truncated and the remaining region (truncated HLR1, tHLR1) was re-synthesized and codon-optimized based on the codon frequency in *Escherichia coli*. The synthetic tHLR1 gene was cloned into the pCzn1 expression vectors, allowing the production of a recombinant tHLR1 fused to a His-tag. Production was performed in the *E. coli* Arcti Express system grown at 37 °C in LB medium. Following incubation and induction, cells were centrifuged and pellets were suspended in a lysis buffer and were sonicated and centrifuged. Crude protein extracts were loaded on a His-Trap HP column (GE Healthcare) and purified. The

purity of the recovered tHLR1 fractions was assayed on SDS-PAGE and concentrated using an Amicon-Ultra device (Millipore). Polyclonal antibodies against tHLR1(anti-HLR1) were raised in two New Zealand White rabbits. Animal experiments were approved by the Animal Care and Use Committee (ACUC; Hainan University) and performed in accordance with the Guidelines of Hainan University for animal experiments. Appropriately, $10^9$ cells were collected by centrifugation and soluble proteins were extracted for SDS-PAGE analysis. Immunoblot analyses were performed as described[58] using the anti-HLR1 antibodies. The signal amplitudes of the bands were quantified by Gel Doc XR System (Bio-Rad).

**Chemical application**. Algal cultures at the linear growth phase (OD750 = 2.5) were harvested by centrifugation and inoculated into the fresh medium with equal numbers. Cells were dark-acclimated overnight, then transferred to 200 μmol photons $m^{-2} s^{-1}$ illumination in the presence or absence of indicated chemicals (e.g., 1 μM DBMIB, 1 mM MV, 255 U $ml^{-1}$ SOD, or 1250 U $ml^{-1}$ CAT). PSII maximum photochemical efficiency was measured after 12 h.

**$H_2O_2$ measurements**. The concentration of $H_2O_2$ was measured by Amplex Red Oxidase Assay Kit Ampliflu (Invitrogen)[59]. Briefly, 1 ml microalgae were taken from cultures at a cell density of $6 \times 10^7$ cells $ml^{-1}$ and immediately supplemented with 5 μM Amplex-Red and 1 U horseradish peroxidase. Cells were removed by centrifugation and the supernatant was measured (excitation, 571 nm; emission, 585 nm). $H_2O_2$ concentrations were determined based on absorption measurements of a standard curve prepared with known concentrations. Measurements were performed on five separate cultures. Samples were taken at 0 h (immediately after overnight dark acclimation), and at 1 h intervals thereafter.

**Transcriptome sampling and analysis**. Cultures were cultured into log phase in continuous light at 50 μmol photons $m^{-2} s^{-1}$ at 25 °C. Mid-logarithmic phase algal cells were transferred into darkness overnight and then exposed to 200 μmol photons $m^{-2} s^{-1}$ for various times. Aliquots of cells for transcript analysis were collected immediately before the light shift, and then 1 and 6 h following the light shift. The total RNA of the algal cells was prepared using an RNA miniprep kit (CWBIO) and the concentrations were measured using the NanoDrop 2000 spectrophotometer (ThermoFisher). Libraries were constructed and then sequenced using the HiSeq 2500 (Illumina), which generated about 18 million read pairs per sample (ANNOROAD). Raw reads containing adapter, poly-N, and low-quality reads were filtered, and the effective data were mapped to the *Nannochloropsis* reference genome using TopHat (version 2.0.12). After excluding the ribosomal RNA or transfer RNA, we estimated the abundance of the transcripts using RPKM (Reads per Kilo bases per Million reads)[60]. The *P* values were adjusted using the Benjamini and Hochberg method[61]. Corrected *P* value < 0.05 and fold change more than two were set as the threshold for a significant difference in expression. GO annotations of the data provided by our RNA-Seq analysis were performed using PANTHER (www.pantherdb.org/pathway/)[62]. Three biologically independent sets of samples were prepared at each time point.

**Statistics and reproducibility**. All statistical tests used are noted in figure legends. All *n* indicated in the figures represent independent experimental samples and not technical replicates. Results are expressed as the mean ± standard deviation (SD). The comparisons between the averages of the two groups were evaluated using the one-tailed Student's *t* test. *P* values of ≤ 0.05 were considered statistically significant. The *P* values with significance are shown in the figures.

**Reporting summary**. Further information on research design is available in the Nature Research Reporting Summary linked to this article.

## Data availability

The data discussed in this publication have been deposited in NCBI's Gene Expression Omnibus[63] and are accessible through GEO Series accession number GSE156681. All other data that support the findings of this study are available from the corresponding authors upon reasonable request. Source data are provided with this paper.

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

## Acknowledgements

We thank Thomas Roach (The French Alternative Energies and Atomic Energy Commission, CEA) for assistance on hydrogen peroxide measurements, Alessandra Bellan (Universita' di Padova) on JTS-10 spectrophotometer analysis, and Eva Farré (Michigan State University) for providing the pNOC-CRISPR vector. We thank Dr. Jian Xu (Qingdao Institute of BioEnergy and Bioprocess Technology, Chinese Academy of Sciences) and Wenqiang Yang (Institute of Botany, Chinese Academy of Sciences) for very helpful discussions. Y.L. and Q.G. gratefully acknowledge funding from the National Natural Science Foundation of China (32060061), the Basic and Applied Basic Research Programs for the Talents of Hainan Province (grant no. 2019RC033), the Project of Innovation & Development of Marine Economy (grant no. HHCL201803), and the Chinese Government Scholarship (grant no. CSC201807565004), the Foundation of Hainan University (grant no. KYQD1561), and the Project of State Key Laboratory of Marine Resource Utilization in South China Sea (grant no. MRUKF2021003). M.I., T.C., and K.K.N. were supported by the U.S. Department of Energy, Office of Science, Basic Energy Sciences, Chemical Sciences, Geosciences, and Biosciences Division under field-work proposal 449B. O.D. was supported by German Research Foundation (DFG) project number 427925948. K.K.N. is an investigator of the Howard Hughes Medical Institute. H.K. was supported by the JSPS KAKENHI Grants (18K06275 and 19H04716) and Network Joint Research Center for Materials and Devices (20196004).

## Author contributions

Y.L. contributed to designing and performing the experiments related to the design and optimization of plasmids for insertional gene disruption, topology predictions, mass spectrometry, DNA gel blotting, gene silencing, and Cas9-mediated gene knockout. Q.G. contributed to performing the experiments related to the gene expression analysis, phenotyping, $H_2O_2$ measurements, and transcriptome sampling and analysis. Y.L. and M.I. contributed to designing and performing the experiments related to the blue-native PAGE and immunodetection. A.A. and T.M. contributed to performing the phylogenetic analysis. Y.L., A.A., T.M., and H.T. contributed to designing and performing the experiments related to the JTS measurements. Y.L., A.B., and G.P. contributed to performing oxygen evolution measurements. Y.L. and O.D. contributed to designing and performing the experiments related to HPLC measurements. Y.L., T.C., and A.M. contributed to designing and performing the experiments related to measurements of light-harvesting antenna size. Y.L. and K.K.N. designed the experiments. Y.L., T.M., A.M., and K.K.N. wrote the paper.

## Competing interests

The authors declare no competing interests.
