## [Peer Review File · Nature Communications]

REVIEWER COMMENTS

Reviewer #1 (Remarks to the Author):

Review of HLR1-LU et al

This paper reports the isolation and characterisation of a *Nannochloropsis* mutant that has increased resistance to high light. Surprisingly, the gene responsible was found to be a member of the Lhcr clade of PSI-associated LHCs rather than the usual photoprotective Lhcx clade. The connection between genotype and phenotype was confirmed with two types of knockouts, and the protein was shown to be associated with PSI macrocomplexes and induced by HL exposure. The mutant had decreased PSI antenna size which resulted in decreased electron flow through PSI and as a result led to lower production of ROS, particularly H₂O₂. This means that the wild-type gene product is actually deleterious under HL while providing increased light-harvesting capacity to PSI under LL. An interesting conundrum.

Most of the experimental work appears to have been very carefully done and supports the conclusions drawn. However, there are some questions that need to be addressed before publication, particularly the transcriptome results.

(1) In Fig. 1b, why was the Chl content normalized on the basis of dry weight, rather than per cell? Especially when there is increased carotenoid content/cell, which could have affected DW.

(2) Fig. 1d. These micrographs are so small as to be unconvincing, even blown up on the screen. Make a separate Supp. figure at a decent magnification. Most readers will believe you and the rest can look it up.

(3) Fig. 2c-The source of the background molecular structure in this panel should be cited. In terms of the relationship with the rest of the LHC family, it would be helpful to show the effect of HL on expression of one or two other members of the Lhcr clade, and a representative Lhcx/LHCSR.

(4) Fig. 5. The transcriptome analysis is the weakest part of the paper. The Methods do not give any detail of how the cells were grown, under exactly what light level, and why such long sampling times were used, especially the 4 and 6 day points. Given the different growth rates of wt and mutant cells (Figure S2), and the fact that the relative growth rates are different under LL and HL, the points at 1 and 6 hr are probably unaffected, but the 96 and 144h points are dubious. The cultures would not be in the same growth phase at the longer sampling times. Were cells transferred to fresh medium during such a long period to keep them in exponential phase? If so, that should be specified.

The cartoon diagram is very useful, but the expression data should have been put in a separate panel, and it should have been explained much more clearly in the Legend. The Legend suggests the heat map is just showing how the wild type genes are regulated in response to transfer from dark to light (irradiance not specified). But the scale in the figure shows it is the two-fold difference in expression between mutant and wt.

The heat map shows that the nitrate transporter and nitrate reductase are only differentially upregulated at the 4 day and 6 day time points. Then the last paragraph of Results switches to considering how WT levels are regulated vs. mutant, i.e. the reverse of what is shown in the figure. This is confusing! And since I'm dubious about the 4 and 6 days time points, and the earlier timepoints are all over the map except for glycosylation, deglycosylation and AA biosynthesis, this part needs to be rethought and rewritten.

The transcriptome results seem to have been added on as an afterthought, perhaps some leftover data from another project. Since the rest of the paper is really quite good and justifies publication, I would suggest showing only the 0, 1 and 6 hours timepoints and adjusting the Results text to match the data. This would not affect the Discussion at all.

Reviewer #2 (Remarks to the Author):

A very nice and thorough study reporting on identification of a new regulatory protein named HLR1 in the red alga *N. oceanica*. The manuscript provide solid evidence for the regulatory role of HLR1, though, rather surprisingly, it is the absence of the protein that provides better protection against high light conditions. Using various spectroscopy and molbio/biochemistry methods, the authors show that absence of the HLR1 protein leads to a decrease of PSI antenna, resulting in less formation of ROS under high light. The manuscript provides important results that deserves to be published. I have a few comments/questions that can be considered during the revision:

1. Did the author measure which carotenoid species are actually present in WT and hlr1 mutant? The change in total carotenoid content (Fig. S1) is rather minor. On the other hand, the alteration of NPQ capacity is significant in the mutant, moreover dependent on prior acclimation to either dark or light (Fig. S6). It seems that zeaxanthin production is somehow altered in the hlr1 mutant, but knowing the complete carotenoid composition in WT and hlr1 mutant could help to better understand the differences in NPQ capacity.
2. I agree with the authors conclusion that the decreased ROS production is likely related to decrease of the cyclic electron flow in hlr1 mutant. Yet, is it actually possible to rule out a scenario, in which the less production of ROS is related to increased efficiency of ROS scavenging (e.g. by carotenoids) in the hlr1 mutant?
3. The authors suggest that HLR1 is likely important during the PSI assembly. In this respect, is there any similarity of HLR1 to cyanobacterial Hlips, which are known to play this role in cyanobacterial PSII? E.g. some conserved AA's. number of pigments per protein...?
4. The reported data show that hlr1 mutant exhibits greater Chl content under high-light condition than WT (Fig. 1b). However, at the same time, the number of Chl per PS1 decreases in hlr1 mutant (Fig. 3b). Thus, if the total Chl content is larger in hlr1 mutant, where is it?

Dear Reviewers,

Thank you for the constructive and extremely valuable comments to help improve our manuscript. We have carefully addressed each of the comments, as detailed below. Here we are submitting a revised version, with item-by-item responses (**bold fonts**) to the reviewers' comments below.

Comments from Reviewer 1:

Most of the experimental work appears to have been very carefully done and supports the conclusions drawn. However, there are some questions that need to be addressed before publication, particularly the transcriptome results.

Response: We greatly appreciate the reviewer's comments, which have helped us to improve the manuscript. We have carefully revised the manuscript to address the remaining scientific issues and improve the readability. Specifically, we added Supplementary Fig. 2 (Representative transmission electron microscopy images of WT and the *hlr1* mutant grown under LL or HL conditions), Supplementary Fig. 6a (Multiple sequence alignment of HLR1 with HLIPs), Supplementary 7b, c, e (Kinetics of violaxanthin and antheraxanthin accumulation determined by time-resolved HPLC measurements and comparison of (A+Z)/(V+ A+Z) values of WT and the *hlr1* mutant), and Supplementary Fig. 10 (Transcriptional dynamics of representative genes encoding LHCRs (a) and LHCXs (b) and genes involved in ascorbic acid biosynthesis (*g10376*; c), cell wall biosynthesis (*g7046*; d), and protein glycosylation (*g9910*; e)). Meanwhile, we removed Supplementary Fig. 1 (Carotenoid content of *N. oceanica* WT and *hlr1* cells cultured under LL or HL conditions) and Supplementary Fig. 10 (Transcript dynamics of *N. oceanica* wild type and *hlr1* cells used for RNA-Seq analysis) of the previous version which are not necessary for the main conclusions of the manuscript. In particular, for the transcriptome analysis, we have excluded 96- and 144-hour time points (including only the 0, 1 and 6-hour timepoints) as suggested by the reviewer. The Results section has been adjusted accordingly. Please refer to our responses below for more details.

1. In Fig. 1b, why was the Chl content normalized on the basis of dry weight, rather than per cell? Especially when there is increased carotenoid content/cell, which could have affected DW.

Response: We greatly appreciate the reviewer's input. We have modified Fig. 1b where Chl content is now normalized on the basis of per cell.

Fig. 1b Chl quantification. The inset shows a comparison of pigment extraction of WT and the *hlr1* mutants under HL conditions.

2. Fig. 1d. These micrographs are so small as to be unconvincing, even blown up on the screen. Make a separate Supp. figure at a decent magnification. Most readers will believe you and the rest can look it up.

Response: As suggested, we have provided Supplementary Fig. 2 to show the micrographs. Now the legend of Supplementary Fig. 2 reads:

“Supplementary Fig. 2. Representative transmission electron microscopy images of WT and the *hlr1* mutant grown under LL or HL conditions. Scale bars are shown. LL, low light ($5 \mu\text{mol}\cdot\text{photons}\cdot\text{m}^{-2}\cdot\text{s}^{-1}$); HL, high light ($200 \mu\text{mol}\cdot\text{photons}\cdot\text{m}^{-2}\cdot\text{s}^{-1}$).”

3. Fig. 2c-The source of the background molecular structure in this panel should be cited.

In terms of the relationship with the rest of the LHC family, it would be helpful to show the effect of HL on expression of one or two other members of the Lhcr clade, and a representative Lhcx/LHCSR.

Response: We greatly appreciate the reviewer’s comments and have clarified the

source of the background molecular structure as suggested. Now the text reads (Line 155 to line 158 in the clean version of the maintext):

“Hidden Markov model-based searches suggest structural similarity of HLR1 to the PSI-LHCI from a green alga *Bryopsis corticulans* (PDB entry 6IGZ in Pfam; <http://pfam.xfam.org/structure/6igz>)¹. HLR1 has 162 residues (81%) modelled with 100.0% confidence by this single template.”

For the expression of other members of LHCR clade, and LHCX/LHCSR, a new Supplementary Fig. 10 has been inserted as below:

Supplementary Fig. 10. Transcript dynamics of *N. oceanica* wild type and *hlr1* cells used for RNA-Seq analysis. (a - b) Transcriptional dynamics of the genes encoding LHCRs (a) and LHCXs (b) in wild-type *N. oceanica* in response to HL (200 $\mu\text{mol}\cdot\text{photons}\cdot\text{m}^{-2}\cdot\text{s}^{-1}$). (c - e) Transcriptional dynamics of genes involved in ascorbic acid biosynthesis (*g10376*; c), cell wall biosynthesis (*g7046*; d), and protein glycosylation (*g9910*; e) in *N. oceanica* wild type and *hlr1* cells in response to HL (200 $\mu\text{mol}\cdot\text{photons}\cdot\text{m}^{-2}\cdot\text{s}^{-1}$).

Correspondingly, new text has been added in the Section “Roles of HLR1 in protection from HL stress” (Line 282 to line 284 in the clean version of the maintext):

“Following the transition from darkness to HL, transcripts of LHCXs (e.g., *g903* and *g1546*; Supplementary Fig. 10a) and LHCRs (*g5629*, *g6113*, *g6882*, and *g9713*; Supplementary Fig. 10b) tend to be increased.”

4. Fig. 5. The transcriptome analysis is the weakest part of the paper. The Methods do not give any detail of how the cells were grown, under exactly what light level, and why such long sampling times were used, especially the 4 and 6 day points. Given the

different growth rates of WT and mutant cells (Figure S2), and the fact that the relative growth rates are different under LL and HL, the points at 1 and 6 hr are probably unaffected, but the 96 and 144h points are dubious. The cultures would not be in the same growth phase at the longer sampling times. Were cells transferred to fresh medium during such a long period to keep them in exponential phase? If so, that should be specified.

The cartoon diagram is very useful, but the expression data should have been put in a separate panel, and it should have been explained much more clearly in the Legend. The Legend suggests the heat map is just showing how the wild type genes are regulated in response to transfer from dark to light (irradiance not specified). But the scale in the figure shows it is the two-fold difference in expression between mutant and wt.

The heat map shows that the nitrate transporter and nitrate reductase are only differentially upregulated at the 4 day and 6 day time points. Then the last paragraph of Results switches to considering how WT levels are regulated vs. mutant, i.e. the reverse of what is shown in the figure. This is confusing! And since I'm dubious about the 4 and 6 days time points, and the earlier timepoints are all over the map except for glycosylation, deglycosylation and AA biosynthesis, this part needs to be rethought and rewritten.

The transcriptome results seem to have been added on as an afterthought, perhaps some leftover data from another project. Since the rest of the paper is really quite good and justifies publication, I would suggest showing only the 0, 1 and 6 hours timepoints and adjusting the Results text to match the data. This would not affect the Discussion at all.

Response: We fully agree with the reviewer and substantially revised the section “Roles of HLR1 in protection from HL stress” (i.e., the transcriptome analysis).

The *Methods* section has been revised to include more details (Line 594 to line 598 in the clean version of the maintext):

“Transcriptome sampling and analysis

Algal cells were cultured into log phase in continuous light at 50 $\mu\text{mol}\cdot\text{photons}\cdot\text{m}^{-2}\cdot\text{s}^{-1}$ at 25 °C. Mid-logarithmic phase cells were transferred into darkness overnight and then exposed to 200 $\mu\text{mol}\cdot\text{photons}\cdot\text{m}^{-2}\cdot\text{s}^{-1}$ for various times. Aliquots of cells for transcript analysis were collected immediately before the light shift, and then 1 and 6 h following the light shift.”

As advised, we have removed 4- and 6-day time points of the transcriptome analysis, deleted the cartoon diagram in Fig. 5, and provided a new version of Supplementary Fig. 10 (Line 893 to line 902 in the clean version of the maintext and Supplementary information).

“Fig. 5. Schematic of electron flow in *N. oceanica*. AA, ascorbic acid; APX, ascorbate peroxidase; b_6f , cytochrome b_6f complex; Cyt c_6 , cytochrome c_6 complex; FD, ferredoxin; FNR, ferredoxin-NADP reductase; L-GalDH, L-galactose dehydrogenase; MDA, monodehydroascorbate; MDAR, monodehydroascorbate reductase; PQ and PQH₂, plastoquinone and reduced plastoquinone; PSII and PSI, photosystem II and I; SOD, superoxide dismutase.

See Supplementary Fig 9 and Supplementary Data 5 and text for details of the transcriptional dynamics of genes involved in cell wall biosynthesis, protein glycosylation, and AA biosynthesis.

Correspondingly, the Results section has been revised (Line 277 to line 282 and Line 294 to line 303):

“Global gene expression of WT and the *hlr1* mutant, as a function of time in dark incubation (DK) and under high irradiance was measured by mRNA-Seq. Cultures were maintained in the dark overnight and then exposed to 200 $\mu\text{mol}\cdot\text{photons}\cdot\text{m}^{-2}\cdot\text{s}^{-1}$ for various times. Aliquots of cells for transcript analysis were collected immediately before the light shift, and then 1 and 6 h following the light shift (see *Methods* for details).

...

Compared with the WT, the L-ascorbic acid (AA) biosynthesis pathway appeared to be decreased in the mutant in several ways: (i) transcripts of the AA biosynthetic gene L-galactose dehydrogenase^{2, 3} (L-GalDH, g10376) were down-regulated (L-GalDH is feedback regulated by AA at the RNA level in some organisms⁴) (Supplementary Fig. 10c); (ii) AA is derived from aldoses such as D-mannose and L-galactose, which can follow several pathways, either toward the biosynthesis of AA, or toward the supply of precursors for cell wall polysaccharide synthesis and protein glycosylation⁵ (Fig. 5). HL suppressed the transcripts of the cellulose biosynthesis gene (i.e., g7046) (Supplementary Fig. 10d) and a gene involved in protein deglycosylation (i.e., g9910) (Supplementary Fig. 10e).”

Please also refer to our response to Comment 3 for details.

Comments from Reviewer 2:

A very nice and thorough study reporting on identification of a new regulatory protein named HLR1 in the red alga *N. oceanica*. The manuscript provides solid evidence for the regulatory role of HLR1, though, rather surprisingly, it is the absence of the protein that provides better protection against high light conditions. Using various

spectroscopy and molbio/biochemistry methods, the authors show that absence of the HLR1 protein leads to a decrease of PSI antenna, resulting in less formation of ROS under high light. The manuscript provides important results that deserves to be published. I have a few comments/questions that can be considered during the revision.

Response: We greatly appreciate the reviewer's comments and have carefully revised as advised. We are sure that these inputs have helped us to improve the manuscript. Please refer to our responses below for more details.

1. Did the author measure which carotenoid species are actually present in WT and *hlr1* mutant? The change in total carotenoid content (Fig. S1) is rather minor. On the other hand, the alteration of NPQ capacity is significant in the mutant, moreover dependent on prior acclimation to either dark or light (Fig. S6). It seems that zeaxanthin production is somehow altered in the *hlr1* mutant, but knowing the complete carotenoid composition in WT and *hlr1* mutant could help to better understand the differences in NPQ capacity.

Response: We fully agree with the reviewer. The *hlr1* mutant shows alteration in NPQ. In LL they have less NPQ than WT (Supplementary Fig. 7a), whereas in HL cells they have more (Supplementary Fig. 7f). This phenotype is very different from that observed in *lhcx1* mutants⁶, consistent with the fact that HLR1 is not homologous to LHCX1. This suggests that the influence of HLR1 is indirect. This indirect effect could be explained by other impacts of the HLR1 mutation such as alteration in carotenoids. As advised, we have investigated the effects of HLR1 on NPQ by measuring the carotenoid composition using time-resolved HPLC. In response to high-light exposure, the *hlr1* mutant has an altered violaxanthin (V; Supplementary Fig. 7b) and antheraxanthin (A; Supplementary Fig. 7c) accumulation. The ability to accumulate zeaxanthin (Z; Supplementary Fig. 7d) was compromised in the *hlr1* mutant and the value of $(A+Z)/(V+A+Z)$ was lower in the mutant than that of WT (Supplementary Fig. 7e).

Accordingly, we have provided a new version of Supplementary Fig. 7 (includes the kinetics of Vio and Anth accumulation) as below:

Supplementary Fig. 7. NPQ properties of wild type (WT) and the *hlr1* mutant. (a) NPQ values in response to dark/high-light exposure of wild type (black circles) and *hlr1* mutant (red squares) acclimated in darkness overnight. Bars at the top of the figures indicate the time sequence of actinic light on (white bar, $200 \mu\text{mol}\cdot\text{photons}\cdot\text{m}^{-2}\cdot\text{s}^{-1}$) and off (black bars). (b) Kinetics of violaxanthin (Vio) accumulation determined by time-resolved HPLC measurements. (c) Kinetics of antheraxanthin (Anth) accumulation determined by time-resolved HPLC measurements. (d) Kinetics of zeaxanthin (Zea) accumulation determined by time-resolved HPLC measurements. (e) Comparison of $(A+Z)/(V+A+Z)$ values of WT and the *hlr1* mutant. V, violaxanthin ; A, antheraxanthin; Z, zeaxanthin. (f) NPQ values in response to dark/high-light exposure of WT (black circles) and *hlr1* mutant (red squares) acclimated in $200 \mu\text{mol}\cdot\text{photons}\cdot\text{m}^{-2}\cdot\text{s}^{-1}$ for 24 h. Data are presented as the means \pm SD ($n \geq 3$).

Moreover, new text has been added in the result section. Now the text reads (Line 180 to line 188):

“To explore whether HLR1 is involved in non-photochemical quenching (NPQ), chlorophyll fluorescence quenching and HPLC analysis of carotenoids in response to dark/light exposure were investigated. The *hlr1* mutant has a slightly lower NPQ capacity (Supplementary Fig. 7a) and a significantly altered violaxanthin (V; Supplementary Fig. 7b) and antheraxanthin (A; Supplementary Fig. 7c) accumulation in response to high-light exposure. The ability to accumulate zeaxanthin (Z; Supplementary Fig. 7d) was compromised in the *hlr1* mutant and the value of $(A+Z)/(V+A+Z)$ was lower in the mutant than that of WT (Supplementary Fig. 7e).”

2. I agree with the authors conclusion that the decreased ROS production is likely related to decrease of the cyclic electron flow in *hlr1* mutant. Yet, is it actually possible to rule out a scenario, in which the less production of ROS is related to increased efficiency of ROS scavenging (e.g. by carotenoids) in the *hlr1* mutant?

Response: We appreciate greatly the inputs here. The ROS production and electron flow were measured using cells in the mid-logarithmic growth phase (6×10^7 cells ml⁻¹). For these cells, no significant difference in carotenoid concentration could be observed between WT and the *hlr1* mutant. Therefore, we assume that, under such a circumstance, potential effects of carotenoids on ROS production could be eliminated. On the other hand, we can't exclude the possibility that, following prolonged HL treatment, relatively higher levels of carotenoids in the *hlr1* mutant (compared with WT) might contribute to the increased efficiency of ROS scavenging. Increased ROS scavenging capacity, together with the compromised biosynthesis of ROS, might jointly lead to the less

production of ROS in the mutant in the long-term HL conditions. To avoid potential confusions, Supplementary Fig. 9 and relating text has been provided in Result section. Now the text reads (Line 263 to line 267):

“On the other hand, following prolonged HL treatment, relatively higher levels of carotenoids in the *hlr1* mutant (compared with WT) was observed (Supplementary Fig. 9). It might contribute to the increased efficiency of ROS scavenging, together with which the compromised biosynthesis of ROS could jointly lead to the less production of ROS in the mutant in the long-term HL conditions.”

3. The authors suggest that HLR1 is likely important during the PSI assembly. In this respect, is there any similarity of HLR1 to cyanobacterial Hlips, which are known to play this role in cyanobacterial PSII? E.g. some conserved AA's. number of pigments per protein...?

Response: We thank the reviewer for this comment. As suggested, we have now provided additional analysis to show similarity of HLR1 to cyanobacterial HLIPs. A new Supplementary Fig. 6a was inserted as below:

Supplementary Fig. 6a. Multiple sequence alignment of HLR1 with high-light-inducible proteins (HLIPs). The folding model of an HLIP of *Synechococcus* sp. strain PCC 7942 is shown. Green box indicates the transmembrane helice in HLIPs and the TM1 in HLR1 or LHCA1. Chl *a*-specific sites are denoted by triangles. Conserved residues are highlighted in red. Abbreviations: HLIP, the HLIP of *Synechococcus* sp. strain PCC 7942 (AAC43401.1); HliA – D, HLIPs of *Synechocystis* sp. strain PCC6803 (ssl2542, ssr2595, ssl1633, and ssr1789); LHCA1, Light-harvesting chlorophyll *a/b*-binding protein of photosystem I of *C. reinhardtii* (Q05093).

```

HLIP  : -ASWGFHDR-----AEKLNGRLAMIGFVALILTEVA----- : 58
HliA  : -VQAGWTKY-----AEKMNGRFAMIGFASLLIMEVV----- : 58
HliB  : -VQAGWTEY-----AEKMNGRFAMIGFVSLAMEVI----- : 58
HliC  : -SKFGFTAF-----AENWNGRLAMIGFSSALILELV----- : 58
HliD  : -PKFGFNNY-----AEKLNGRAAMVGFLLILVIEYF----- : 45
LHCA1 : PGNYGFDPLSLGKEPASLKRFTSEVIHGRWAMLGVAGSLAVELLGYGNWYDAPL : 108
HLR1  : AGDFGFDPMGISDQVANLKYVRAAEELKHCRVAMLGFLGWVVQQYVHLPG----EI : 105

```

▲ ▲ ▲

Correspondingly, new text has been added in Section “The *hlr1* mutant is defective in a HL-induced *LHCR* gene”. Now the text reads (Line 160 to line 164):

“Like other LHCs, the TM1 of HLR1 shares sequence similarity to

cyanobacterial HLIPs (single transmembrane polypeptide ⁷) with conserved Chl *a* binding sites (Supplementary Fig. 6a), supporting that LHC polypeptides may have arisen by a gene duplication of an ancestral gene encoding a single TMH polypeptide ⁸”.

4. The reported data show that *hlr1* mutant exhibits greater Chl content under high-light condition than WT (Fig. 1b). However, at the same time, the number of Chl per PS1 decreases in *hlr1* mutant (Fig. 3b). Thus, if the total Chl content is larger in *hlr1* mutant, where is it?

Response: We apologize for not explaining this clearly. While there was less Chl per PSI in the mutant, the absorption cross section of PSII increased, so we believe the extra Chl is associated with PSII. We have tried to clarify this point in the text. Now the text reads (Line 328 to line 337):

“Using log-phase microalgae, we observed a moderately larger absorption cross section σ (PSII) (about 17%) in the mutant compared to the WT, suggesting a higher PSII antenna size. Meanwhile, the mutant showed lower Chl content per P700 (per electron transport chain) than WT. Overall, in these cells, no significant difference could be observed in Chl content between WT and the mutant. On the other hand, under the high-light condition, the *hlr1* mutant exhibits greater Chl content and a higher maximal quantum yield of PSII, affording evidence that the Chl content in PSII might be higher in the mutant. Thus, higher Chl content in PSII might be a contributor to the larger total Chl content in the *hlr1* mutant in HL conditions.”

Again, thank you for the constructive comments that have improved our manuscript greatly.

Sincerely yours,

Yandu LU, Ph.D
College of Oceanology
Hainan University
Email: ydlu@hainanu.edu.cn
<http://orcid.org/0000-0002-0136-2252>

Krishna K. Niyogi
Investigator, Howard Hughes Medical Institute
Professor, Department of Plant and Microbial Biology, University of California
Berkeley
Faculty Scientist, Lawrence Berkeley National Laboratory
Laboratory webpage: <http://niyogilab.berkeley.edu/>

References:

1. Qin, X. et al. Structure of a green algal photosystem I in complex with a large number of light-harvesting complex I subunits. *Nature Plants* **5**, 263-272 (2019).
2. Gatzek, S., Wheeler, G.L. & Smirnov, N. Antisense suppression of l-galactose dehydrogenase in *Arabidopsis thaliana* provides evidence for its role in ascorbate synthesis and reveals light modulated l-galactose synthesis. *The Plant Journal* **30**, 541-553 (2002).
3. Wheeler, G.L., Jones, M.A. & Smirnov, N. The biosynthetic pathway of vitamin C in higher plants. *Nature* **393**, 365-369 (1998).
4. Mieda, T. et al. Feedback inhibition of spinach L-galactose dehydrogenase by L-ascorbate. *Plant and Cell Physiology* **45**, 1271-1279 (2004).
5. Smirnov, N., Conklin, P.L. & Loewus, F.A. Biosynthesis of ascorbic acid in plants: A Renaissance. *Annual Review of Plant Biology* **52**, 437-467 (2001).
6. Park, S. et al. Chlorophyll–carotenoid excitation energy transfer and charge transfer in *Nannochloropsis oceanica* for the regulation of photosynthesis. *Proceedings of the National Academy of Sciences* **116**, 3385-3390 (2019).
7. Dolganov, N.A., Bhaya, D. & Grossman, A.R. Cyanobacterial protein with similarity to the chlorophyll a/b binding proteins of higher plants: evolution and regulation. *Proceedings of the National Academy of Sciences* **92**, 636-640 (1995).
8. Kühlbrandt, W., Wang, D.N. & Fujiyoshi, Y. Atomic model of plant light-harvesting complex by electron crystallography. *Nature* **367**, 614-621 (1994).

REVIEWERS' COMMENTS

Reviewer #1 (Remarks to the Author):

The authors have extensively revised their manuscript in the light of my comments and those of the other reviewers. It is now quite a good paper, and worthy of publication in your journal.

Reviewer #2 (Remarks to the Author):

Then authors have responded to all my comments and clarified some issues I have indicated in my comments. I believe the revised manuscript is suitable for publication.

Tomas Polivka